# Remote sensing quantifies widespread abundance of permafrost region disturbances across the Arctic and Subarctic

I. Nitze [1], G. Grosse [1,2], B. M. Jones[3], V.E. Romanovsky[4,5] & J. Boike [1,6]

Local observations indicate that climate change and shifting disturbance regimes are causing permafrost degradation. However, the occurrence and distribution of permafrost region disturbances (PRDs) remain poorly resolved across the Arctic and Subarctic. Here we quantify the abundance and distribution of three primary PRDs using time-series analysis of 30-m resolution Landsat imagery from 1999 to 2014. Our dataset spans four continental-scale transects in North America and Eurasia, covering ~10% of the permafrost region. Lake area loss ($-1.45\%$) dominated the study domain with enhanced losses occurring at the boundary between discontinuous and continuous permafrost regions. Fires were the most extensive PRD across boreal regions (6.59%), but in tundra regions (0.63%) limited to Alaska. Retrogressive thaw slumps were abundant but highly localized ($<10^{-5}\%$). Our analysis synergizes the global-scale importance of PRDs. The findings highlight the need to include PRDs in next-generation land surface models to project the permafrost carbon feedback.

[1] Alfred Wegener Institute, Helmholtz Centre for Polar and Marine Research, 14473 Potsdam, Germany. [2] Institute of Earth and Environmental Science, University of Potsdam, 14476 Potsdam, Germany. [3] Institute of Northern Engineering, University of Alaska Fairbanks, Fairbanks, Alaska 99775, USA. [4] Geophysical Institute, University of Alaska Fairbanks, Fairbanks, Alaska 99775, USA. [5] Department of Cryosophy, Tyumen State University, Tyumen, Russian Federation, 625000. [6] Geography Department, Humboldt-Universität zu Berlin, 10099 Berlin, Germany. Correspondence and requests for materials should be addressed to I.N. (email: ingmar.nitze@awi.de)

Climate change and disturbance regime shifts are amplified in the northern high latitudes (>60° N), and global and regional projections indicate that some environmental thresholds linked to Earth's Cryosphere, such as the loss of permanent summer sea ice cover in the Arctic Ocean, will be crossed during the twenty-first century (IPCC, RPC 8.5)[1]. Permafrost, which affects roughly 24% of land surface of the northern hemisphere, is an important component of the Cryosphere. It is warming in response to these changes[2,3] and 50−90% of near surface permafrost in Arctic and Boreal regions are projected to be lost by 2100[4,5]. Widespread loss of near surface permafrost will mobilize and release a large reservoir of perennially frozen soil carbon to the atmosphere[6,7], which will have ramifications for Earth's climate system[8]. Increased carbon emissions from thawing permafrost may thus further enhance warming temperatures, a process known as the permafrost carbon feedback[9]. A recent synthesis study combining spatial datasets of permafrost, soil carbon, and terrain with an empiric classification scheme indicates that landscapes vulnerable to rapid thaw processes contain a major portion of the permafrost-stored soil carbon[10]. Due to rapidly changing climate and increased local anthropogenic influences from economic development in high northern latitude regions, permafrost increasingly shows signs of rapid degradation in many Arctic and Boreal regions[11–17]. Rapid land surface dynamics in the permafrost region are indicators (lake changes, mass wasting processes) and/or triggers (fire) of permafrost degradation and potentially amplify the intensity and velocity of permafrost degradation. However, spatially and temporally consistent inventories of permafrost-affecting local landscape changes across very large regions in sufficiently high spatial-resolution are currently lacking.

The northern permafrost region (Fig. 1) varies with respect to spatial extent and characteristics of ground thermal regime, ground-ice content, climate, topography, hydrology, surface geology, and land cover[10,18–20]. These factors interact in complex ways such that predicting the response of permafrost terrains to climate change and disturbances beyond local scales is extremely difficult[21]. For example, changes in climate and disturbance regimes may trigger an increase or a decrease in thermokarst lake numbers[16,22]. They may also cause widespread ice-wedge degradation or promote stabilization through paludification[13,23,24]. Various studies tried to link lake changes to local or regional permafrost extent[25,26]. In other regions, changes in climate and disturbance regimes may initiate retrogressive thaw slumps (RTS) or detachment slides or promote regional stabilization of currently active erosion features[7,15,27]. In addition, with climate warming northern high latitude fire regimes are expected to shift towards shorter fire return intervals, increased burn severity, and more widespread occurrence[28,29]. Fire events, which are already frequent in boreal regions[30–32] but scarcely studied in tundra[33,34], have the ability to initiate or strengthen permafrost disturbances depending on fire severity and timing[35]. While thermokarst lakes and RTS are direct indicators of permafrost disturbances, linkages between wildfires and permafrost degradation are indirect. Wildfires are potential instigators of permafrost degradation following only years after the fire event. The occurrence and magnitude of resulting permafrost disturbance depends on burn severity and subsequent changes in soil surface thermal properties due to loss or severe degradation of vegetation and organic layers[35,36]. Feedbacks and local-scale consequences may trigger widespread changes in permafrost-related processes, causing the mobilization and potential release of carbon to the atmosphere[7] as well as a wide range of ecological and hydrological impacts that remain poorly documented[37].

The occurrence and spatial distribution of contemporary permafrost region disturbances (PRDs) are poorly represented in global assessments due to the previously unresolved conflict of scales between a diverse suite of rapid local-scale processes that sometimes have a very high abundance on the landscape and coarse-resolution continental-scale remote sensing data and processing capabilities. The vast majority of permafrost areas are underrepresented in global remote sensing and modeling studies and most PRDs likely remain undocumented, resulting in significant uncertainty of the current magnitude of rapid permafrost degradation processes and their role in global scale biogeochemical dynamics[7]. This fundamental knowledge gap raises the question: How extensive are recent PRDs and do they vary by permafrost extent and characteristics across the Arctic and Subarctic?

With growing archives of freely accessible earth observation data of adequate spatial scale for permafrost remote sensing as well as rapidly growing computational processing capacities, we now have the potential tools to detect and observe widely distributed PRDs in high spatial and temporal resolution across very large regions. In this study, we analyzed 16 years of 30-m resolution earth observation data from the Landsat archive in conjunction with Pan-Arctic to global scale data products, from 1999 through 2015, to track key PRDs (lake extents and their dynamics, RTS, and wildfire burn scars; Fig. 2) across four extensive latitudinal transects in Alaska, Eastern Canada, Western Siberia, and Eastern Siberia that cover more than $2.3 \times 10^6$ km$^2$ (~10% of the permafrost region in the Northern Hemisphere) (see Supplementary Note 1). Our study domain features a broad range of permafrost, climate, topographic, and geo-ecological zones (Fig. 1). We combined temporal trend-analysis with machine-learning to map the spatial distribution of PRDs and their relation to permafrost properties, ecological zones and climate, allowing comprehensive and unique insights into PRD distribution, abundance, and dynamics as well as permafrost vulnerability to thaw. Our results provide a baseline for improving future landscape models and carbon emission estimations from rapidly evolving PRDs.

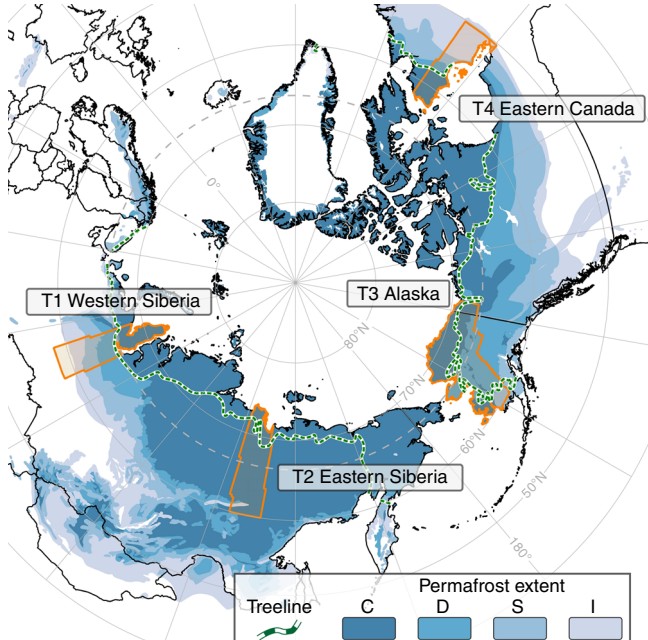

**Fig. 1** Overview of study regions. Four continental-scale permafrost region disturbance (PRD) change detection study areas (orange polygons) overlain on a circumpolar permafrost extent map[20] and treeline delineation[19]. C: continuous permafrost, D: discontinuous permafrost, S: sporadic permafrost, I: isolated permafrost

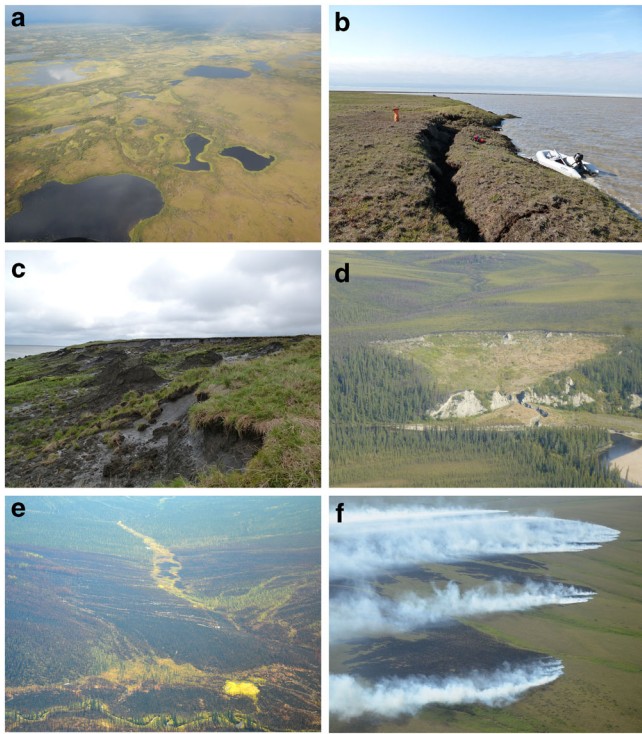

**Fig. 2** Examples of key permafrost region disturbances. **a** Dynamic lake-rich region in western Alaska with frequent drainage, **b** Expanding thermokarst lake in northern Alaska, **c** Coastal retrogressive thaw slump on Bykovsky Peninsula in northeastern Siberia, **d** Selawik thaw slump in western Alaska, **e** Burn scar of wildfire in boreal Alaska, and **f** Burning tundra fire in northern Alaska. Photos taken by I. Nitze (**b**−**d**), B.M. Jones (**e**, **f**), and M. Fuchs (**a**)

## Results and Discussion

**Lakes**. We detected 643,304 lakes with a size larger than 1 ha, with a total area of 118,182 km$^2$ in 2014 or 5.1% of the analyzed land area (Table 1). The lake distribution differed significantly between Eastern Canada (T4), where lakes are highly abundant (13.4%) and distributed homogeneously, and the remaining regions, which have a lower overall limnicity (fraction of lake area) (T1-Western Siberia: 6.1%, T2-Eastern Siberia: 1.6%, T3-Alaska: 2.9%) and a scattered spatial distribution of several very lake-rich spatial clusters (Fig. 3). Lakes in Eastern Canada are primarily located in glacially carved depressions on the bedrock surface of the formerly glaciated Canadian Shield, whereas lakes in the Alaskan and Siberian sites are predominantly located in ice-rich permafrost-affected sediments. Clusters of high limnicity (>15% water) are generally located in flat coastal lowlands (e.g. Alaska North Slope), river valleys and deltas (e.g. Yukon-Kuskokwim, Lena) or interior basins (e.g. Old Crow Flats) with predominantly ice-rich sediments and continuous permafrost and are primarily of thermokarst or fluvial origin (Fig. 3) . However, wetland areas at the fringes of the permafrost zone (e.g. West Siberian Peatland) also form dense lake clusters. Lake sizes are statistically similar for all sites with long-tailed (exponential) distributions and mean lake sizes of 2.96−3.87 ha.

Observed lake changes from 1999 to 2014 were highly diverse in the Alaskan and the two Siberian transects with a wide range from stability to rapid high magnitude changes, which aligned with the heterogeneous spatial patterns of surface geology, geomorphology, permafrost extent, and ground-ice conditions (Fig. 3, Table 1). In contrast, the spatial dynamics of lakes in the Eastern Canadian Region were coherent with the

geomorphological homogeneity and followed a latitudinal gradient of intensifying lake area loss from south to north. Overall lake area loss outweighed lake area gain, particularly in western Siberia with a net change of −5.46% (gross increase and gross decrease in brackets hereafter; +1.58; −7.04%) as well as in Alaska and eastern Canada with net changes of −0.62% (+3.31%; −3.94%) and −0.24% (+1.87%, −2.12%), respectively. The East Siberian transect is characterized by a positive lake area trend with a net change of +3.67% (+7.77%; −4.10%). Overall, gross lake area loss totaled 4767 km$^2$ (−3.98%), whereas gross lake growth accounted for 3030 km$^2$ (2.53%) leading to a net loss of 1737 km$^2$ (−1.44%) or a change from 119,918 km$^2$ to 118,182 km$^2$ total lake area in the period from 1999 to 2014.

Lake area loss is the dominant lake-related process in discontinuous permafrost regions and along the continuous-to-discontinuous permafrost boundary. In discontinuous permafrost regions in Western Siberia and Alaska there was a net lake loss of 7.89 and 5.96%, respectively. Intensive gross lake area loss, e.g. through drainage and/or drying, was the key driver of negative lake area balance, particularly in the Alaska discontinuous permafrost region, with 11.39% gross loss and a simultaneous 5.43% gross gain, which signifies the rapid lake dynamics in this region with drainage on the one hand and lake expansion on the other. Lake districts in this region, e.g. Yukon Flats, Kobuk-Selawik Lowlands or northern Seward Peninsula were among the most dynamic regions of lake change with dominating lake area loss and most are located along the transition zone between continuous and discontinuous permafrost. In Western Siberia, zones of strong lake area loss extended from the discontinuous into the continuous permafrost zone, where a large cluster of lakes on the southern and southeastern Yamal peninsula was particularly affected by partial drainage of large lakes (>10 km$^2$) (Supplementary Fig. 1). The pronounced lake area loss contrasts earlier findings of lake area expansion in continuous permafrost, but supports widespread lake area loss in discontinuous permafrost in this region between 1997 and 2004[25]. For the other two transects no such relationship was observed since the eastern Siberian transect lies nearly completely within continuous permafrost, whereas the eastern Canadian transect is characterized by bedrock geology, where lakes of non-thermokarst origin dominate but still have a thermal impact on the underlying bedrock permafrost.

Lake changes were highly diverse in the continuous permafrost zone. Widespread lake expansion on a massive scale took place on the eastern banks of the Lena River in central Yakutia in ice-rich thick continuous permafrost, where lake area increased by 50% within a short time period (Supplementary Fig. 1). Lake area loss dominated the continuous permafrost section of western Siberia with −4.29% (+1.70%; −5.99%), with increasing lake stability towards the north. The continuous permafrost zone in Alaska had a diverse pattern of localized lake change, with regions of intensive lake dynamics but little net change (North Slope Outer Coastal Plain, YK-Delta), regions with nearly stable conditions (North Slope Inner Coastal Plain), or strong lake area loss (Northern Seward Peninsula). Overall, lake change is nearly evenly distributed between lake area gain and loss with a small overall lake area loss of −0.24% (+3.21%, −3.45%).

Outside the continuous-to-discontinuous permafrost transition zone, permafrost extent did not have a general influence on the net direction and magnitude of (thermokarst) lake changes. Gross lake growth consistently increased towards continuous permafrost in both regions with variable permafrost extent (Western Siberia, Alaska). However, the much more variable lake area loss rates outweighed lake growth rates and determined the net lake

**Table 1 Regional lake area and lake change statistics per study site**

| | | Unit | Overall | Permafrost extent | | | | Ice content | | |
|---|---|---|---|---|---|---|---|---|---|---|
| | | | | C | D | S | I | High | Medium | Low |
| T1 | Net | km² | −1759.50 | −556.19 | −390.11 | −155.99 | −300.98 | −636.69 | −347.67 | −418.89 |
| | | % | −5.46 | −4.29 | −7.89 | −1.94 | −13.58 | −3.67 | −5.35 | −9.70 |
| | Growth | km² | 509.93 | 220.93 | 58.24 | 87.86 | 114.55 | 324.59 | 108.00 | 48.99 |
| | | % | 1.58 | 1.70 | 1.18 | 1.09 | 5.17 | 1.87 | 1.66 | 1.13 |
| | Loss | km² | 2269.42 | 777.12 | 448.35 | 243.84 | 415.52 | 961.28 | 455.67 | 467.88 |
| | | % | 7.04 | 5.99 | 9.07 | 3.04 | 3.04 | 5.54 | 7.01 | 10.84 |
| | *n* Lakes | # | 218,882 | 97,723 | 45,070 | 57,144 | 10,049 | 117,572 | 55,021 | 37,393 |
| | Lake area (2014) | km² | 30,456.53 | 12,422.21 | 4551.18 | 7869.90 | 1915.71 | 16,706.28 | 6154.88 | 3897.84 |
| | | % | 6.10 | 8.38 | 4.29 | 9.73 | 2.55 | 7.99 | 6.28 | 3.77 |
| T2 | Net | km² | 313.81 | 320.08 | −0.02 | 0.00 | 0.00 | 32.00 | 97.82 | 190.65 |
| | | % | 3.67 | 3.81 | −3.12 | 0.00 | 0.00 | 0.55 | 8.30 | 13.99 |
| | Growth | km² | 663.76 | 657.23 | 0.02 | 0.00 | 0.00 | 248.28 | 147.56 | 262.11 |
| | | % | 7.77 | 7.83 | 3.16 | 0.00 | 0.00 | 4.24 | 12.52 | 19.24 |
| | Loss | km² | 349.96 | 337.14 | 0.04 | 0.00 | 0.00 | 216.28 | 49.74 | 71.46 |
| | | % | 4.10 | 4.02 | 6.28 | 0.00 | 0.00 | 3.69 | 4.22 | 5.25 |
| | *n* Lakes | # | 69,151 | 67,156 | 19 | 0 | 0 | 44,058 | 10,670 | 12,579 |
| | Lake area (2014) | km² | 8856.45 | 8714.04 | 0.54 | 0.00 | 0.00 | 5890.20 | 1276.44 | 1553.08 |
| | | % | 1.60 | 1.61 | 0.01 | 0.00 | 0.00 | 2.87 | 1.41 | 0.61 |
| T3 | Net | km² | −161.45 | −55.61 | −109.08 | 3.20 | 0.00 | −123.32 | 43.21 | −81.38 |
| | | % | −0.62 | −0.24 | −5.96 | 0.28 | 0.00 | −1.12 | 0.34 | −3.53 |
| | Growth | km² | 862.19 | 736.96 | 99.40 | 20.92 | 0.00 | 105.48 | 672.12 | 79.68 |
| | | % | 3.31 | 3.21 | 5.43 | 1.83 | 0.00 | 0.96 | 5.34 | 3.45 |
| | Loss | km² | 1023.64 | 792.57 | 208.48 | 17.72 | 0.00 | 228.81 | 628.91 | 161.06 |
| | | % | 3.94 | 3.45 | 11.39 | 1.55 | 0.00 | 2.07 | 5.00 | 6.98 |
| | *n* Lakes | # | 158,453 | 133,813 | 20,219 | 3358 | 0 | 51,279 | 94,101 | 12,010 |
| | Lake area (2014) | km² | 25,851.69 | 22,910.72 | 1721.51 | 1147.63 | 0.00 | 10,920.92 | 12,632.36 | 2226.58 |
| | | % | 2.88 | 4.36 | 0.54 | 2.30 | 0.00 | 11.07 | 2.44 | 0.80 |
| T4 | Net | km² | −129.77 | −438.72 | −82.18 | −52.62 | 447.60 | −2.91 | −0.27 | −122.74 |
| | | % | −0.24 | −2.02 | −1.50 | −1.07 | 2.15 | −1.32 | −1.55 | −0.23 |
| | Growth | km² | 994.59 | 167.69 | 26.43 | 22.51 | 774.63 | 2.02 | 0.18 | 989.06 |
| | | % | 1.87 | 0.77 | 0.48 | 0.46 | 3.71 | 0.91 | 1.03 | 1.87 |
| | Loss | km² | 1124.36 | 606.41 | 108.61 | 75.13 | 327.03 | 4.93 | 0.46 | 1111.80 |
| | | % | 2.12 | 2.79 | 1.99 | 1.53 | 1.53 | 2.23 | 2.59 | 2.11 |
| | *n* Lakes | # | 196,818 | 97,742 | 22,311 | 19,470 | 55,226 | 1506 | 249 | 192,994 |
| | Lake area (2014) | km² | 53,016.88 | 21,322.34 | 5381.98 | 4858.19 | 21,311.32 | 218.50 | 17.39 | 52,637.93 |
| | | % | 13.44 | 15.46 | 17.52 | 15.02 | 11.10 | 4.34 | 4.48 | 13.58 |

Regional lake area (end of observation period in 2014) and lake change statistics per transect as net change, gross lake area growth and gross lake area loss, subdivided by permafrost extent, and ice content. Differences of summed up values to overall area are due to regions of permafrost absence and minor spatial dataset inconsistencies
C: continuous permafrost, D: discontinuous permafrost, S: sporadic permafrost, I: isolated permafrost

change budget (Fig. 4, Table 1). We could not find a general and consistent correlation of ground-ice content with the magnitude and direction of lake changes in contrast to lake abundance, which correlates strongly with ground-ice content (Fig. 4, Table 1).

The reduced susceptibility of lakes to lateral expansion and the relative landscape homogeneity of the Eastern Canadian transect were reflected in suppressed gross lake growth rates, when compared to the Siberian and Alaskan transects which are characterized by abundant thermokarst lakes that expand gradually through shoreline thermo-erosion processes. Both, gross lake growth and loss, gradually increased from south to north along a latitudinal gradient. Strong lake area loss in the northern and northwestern coastal and near-coastal zone of the Eastern Canadian transect outweighed marginal lake growth leading to a net lake area loss of −2.02% (+0.77%; −2.79%) in the northern continuous permafrost zone, which has also been identified in earlier MODIS-based studies[38]. The central and southern portions of the transect show little change with a few clusters of lake growth. The spike in lake area gain at 53–54° N was caused by the filling of the Eastmain-1A reservoir (+470

km²) (Supplementary Fig. 1), which is part of a larger chain of hydro-electrical dams.

Due to the mostly non-thermokarst origin of lakes, lake change in this region did not correlate with permafrost extent and ground-ice datasets. The gradual shift in lake change patterns along a latitudinal gradient thus most likely indicates an influence of large-scale climatic patterns across the transect. The entire transect was affected by an increase of mean annual air temperatures (+1.2 to +1.7 °C), in comparison to 1979 through 1998, in conjunction with a marginal increase in precipitation north of 56°N (+1 to +18 mm) and a stronger precipitation increase in the southern part (+25 to +63 mm) for the observation period (Supplementary Fig. 2).

**Wildfires**. Wildfire burn scars for the 2000–2015 observation period (see Methods section for description of shift in observation period) were widespread in all transects, particularly in the forested boreal region, which we define as limited by the treeline after Walker et al.[19] Southern inland locations with continental climatic conditions characterized by high annual temperature

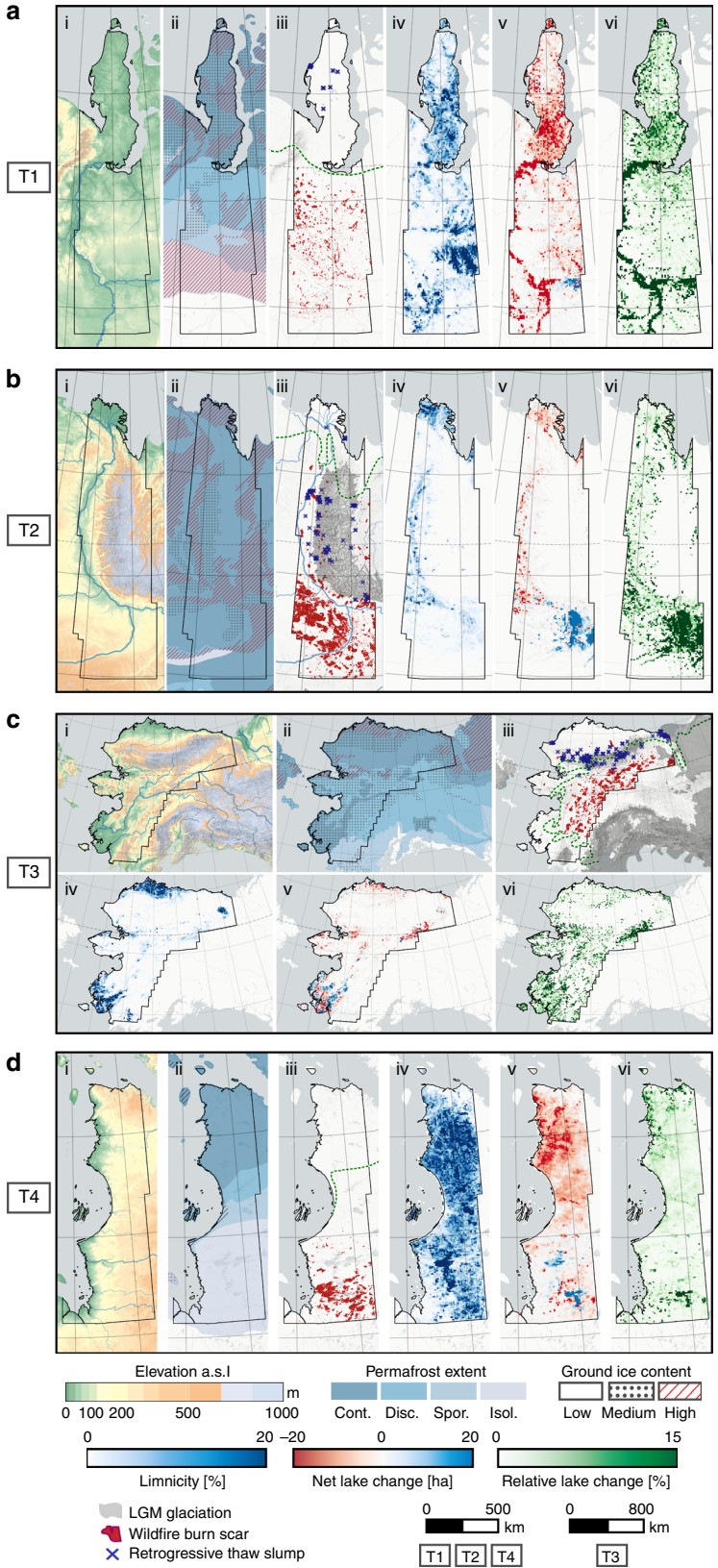

**Fig. 3** Environmental conditions and results of permafrost regions disturbance detection. Lake statistics and permafrost conditions in the four continental-scale, latitudinal transects: **a** T1 Western Siberia, **b** T2 Eastern Siberia, **c** T3 Alaska, and **d** T4 Eastern Canada. (i) Surface elevation of study sites, (ii) permafrost conditions with permafrost extent in shades of blue and ice content based on IPA Permafrost map[20], (iii) Wildfire burn scars, retrogressive thaw slumps, LGM glaciation extent[63] and treeline[20]. LGM glacial coverage omitted in T4 for visual purposes, (iv) Limnicity: Lake fraction of land surface per grid cell (7.5 ×7.5 km), (v) Net lake change: lake area change per grid cell (7.5 ×7.5 km), (vi) Relative lake change: fraction of changing lake area to stable lake area

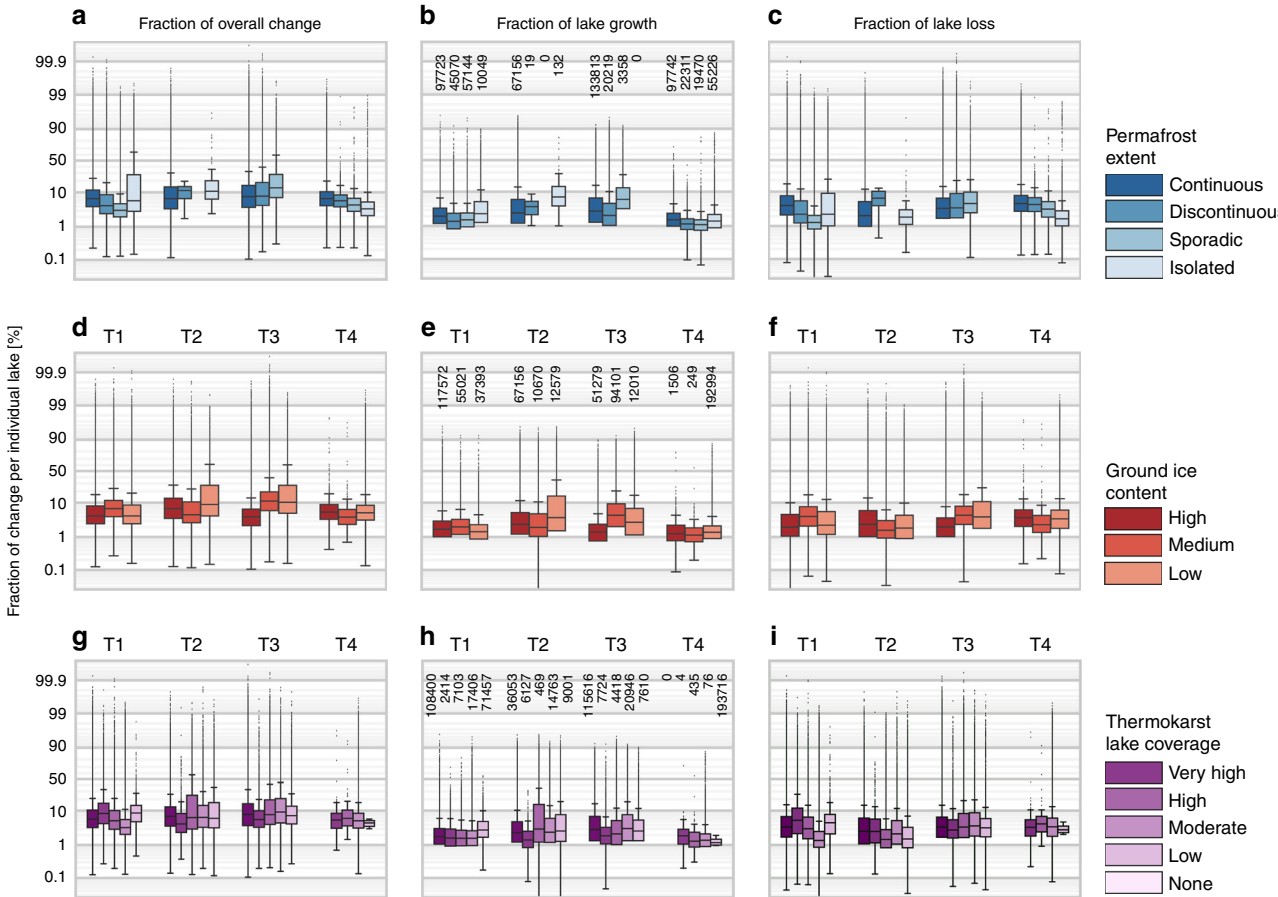

**Fig. 4** Lake change and its relation to permafrost properties. Boxplots of individual lake change statistics per study area (columns) to permafrost properties (rows). **a**−**c** Overall change (**a**), gross lake growth (**b**), and gross lake loss (**c**) by permafrost extent[20]; **d**−**f** overall change (**d**), gross lake growth (**e**), and gross lake loss (**f**) by ground-ice content[20], **g**−**i** overall change (**g**), gross lake growth (**h**), and gross lake loss (**i**) by thermokarst lake coverage[10]. Sample sizes per group (*n* lakes) indicated in middle column (**b**, **e**, **h**). Note: *y*-axes in logit scale. For each boxplot the centerline indicates the median, the box indicates the interquartile range (IQR), and the whiskers indicate 1.5 times the IQR past the low and high quartiles (Tukey boxplot)

## Table 2 Regional statistics of wildfires and retrogressive thaw slumps per study site

| | | Wildfire | | | Retrogressive thaw slumps | |
|---|---|---|---|---|---|---|
| Transect | Unit | Overall | Boreal | Tundra | Unit | Overall |
| T1 | km² | 8156.78 | 8144.62 | 12.16 | km² | 0.21 |
| | % | 1.63 | 2.34 | 0.01 | # | 23 |
| T2 | km² | 42,972.60 | 42,972.60 | — | km² | 1.08 |
| | % | 7.77 | 8.14 | — | # | 140 |
| T3 | km² | 46,162.35 | 41,580.53 | 4581.82 | km² | 4.00 |
| | % | 5.14 | 8.89 | 1.07 | # | 245 |
| T4 | km² | 13,668.59 | 13,668.59 | — | km² | — |
| | % | 3.46 | 5.07 | — | # | — |
| SUM | km² | 110,960.31 | 106,366.33 | 4581.82 | km² | 5.29 |
| | % | 4.73 | 6.59 | 0.63 | # | 408 |

Statistics of areal extent and percentage of wildfire burn scars per transect with distinction of boreal and tundra fires as well as areal extent and number of retrogressive thaw slumps per transect

amplitudes and dry conditions throughout the year had a higher abundance of fires (Table 2).

Eastern Siberia was among the most strongly affected region, where around 7.77% of the total area and 8.14% of the boreal area was burned between 2000 and 2015 (Fig. 3, Table 2). In its southernmost region west of the Lena River, around 17% of the land surface was affected by wildfires, predominantly during a few severe fire seasons[39]. Wildfires also affected large swaths of boreal Alaska with 8.89% burned area (Fig. 3), correlating with drying conditions over the observation period (Supplementary Fig. 2). In the Eastern Canadian Transect, fires were a common occurrence in the southern boreal region where they affected 5.07% of the area. North of 53° latitude wildfire occurrence decreased sharply. In more humid and wetland-dominated Western Siberia, fires were less widespread with a total extent of 1.63% across the entire transect or 2.34% in the boreal area (Fig. 3).

Tundra fires occurred infrequently and were mostly limited to a small extent of 0.63% overall. Alaska stands out as the only region with a noticeable amount of detected tundra fires over the observation period, most prominently the extensive Anaktuvuk Fire in northern Alaska in 2007, which affected an area of ~1000 km². Overall, tundra fires in northern and western Alaska burned nearly 4600 km² or 1.07% of the Alaskan tundra region. In Western Siberia tundra fires affected an area of 12.16 km² or 0.01%.

**Retrogressive thaw slumps**. Actively expanding or newly initiated RTS and landslides larger than 0.1 ha were identified in sloped terrain in the foothills of mountain ranges, as well as

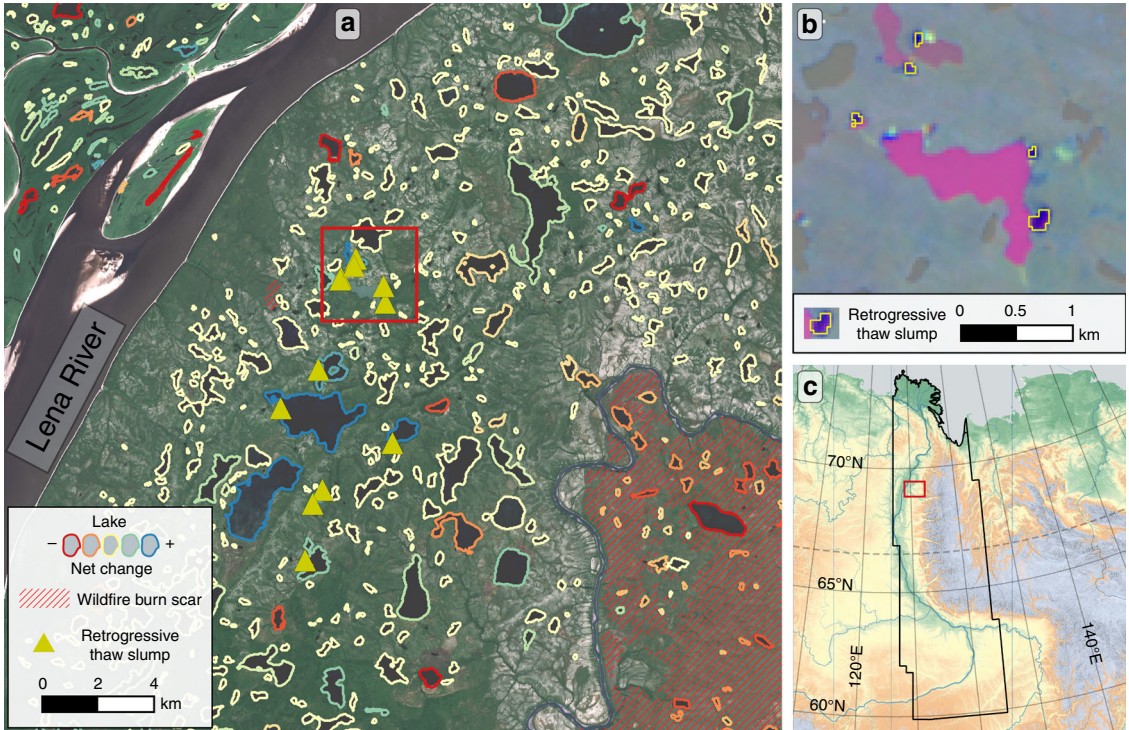

**Fig. 5** Local example of lake changes, retrogressive thaw slumps, and wildfire burn scars along the Lena River in northeastern Siberia. **a** Lake perimeters with net change direction and location of retrogressive thaw slumps (RTS) and wildfire burn scars, **b** Detailed view of Landsat trend images with RTS perimeters indicated along lake shores. Magenta color indicates strong increase in suspended sediments within the lake, **c** Location of example within East Siberian Transect. Approximate Location: 69.13°N, 124.45°E. Background image **a** Sentinel-2 image acquired on July 2, 2018. Background image **b** Landsat Trend Analysis Tasseled Cap Visual Product[64]. Background image **c** GTOPO30 DEM overlain by MapSurfer ASTER GDEM-SRTM Hillshade layer accessed through QGIS QuickMapServices plugin

coastal bluffs, lake shores or valleys in ice-rich permafrost terrain. They typically formed regional clusters of up to 25 individual RTS, within the detection limit, and they were limited to the continuous permafrost zone (Fig. 3).

On the western and central Yamal Peninsula, we identified 23 RTS which are distributed in spatial clusters along the western coast and in the vicinity of the Bovanenkovo gas field. In the Lena Delta region of the east Siberian Transect, a cluster of eight RTS were detected on steep coastal bluffs on the Bykovsky Peninsula south-east of the Lena Delta, where very ice-rich Yedoma Ice-Complex sediments are actively eroded (Fig. 2c). Several isolated clusters, with a total of 115 active RTS were detected in the western and southern foreland of the Verkhoyansk mountain range (VMR), in most cases along lake shores in hummocky terrain associated with glacial moraines of pre-Holocene glaciation lobes protruding from VMR (Fig. 5). In the eastern forelands of the VMR, only 18 RTS were identified. These RTS in the vicinity of the VMR are most likely caused by permafrost degradation associated with decaying glacial ice buried in moraines (Fig. 5). In the Alaskan transect several clusters of RTS and landslides were detected within the Brooks Range and along its northern and western foothills ($n = 184$), most notably in the upper Noatak valley ($n = 52$). Furthermore, 31 coastal RTS were observed on the formerly glaciated Herschel Island and Yukon coast (Canada), whereas another cluster of 9 RTS was detected on steep coastal bluffs on the northwestern coast of Alaska. The large Selawik Slump (Fig. 2d, Supplementary Fig. 1) in western Alaska was detected, but it remains a singular feature in its vicinity and marks the southernmost detected RTS in the Alaskan Transect. No actively eroding RTS were detected in the eastern Canadian Transect.

**Implications for Arctic and Boreal regions**. The analyzed PRDs can be grouped into spatially extensive (lakes, fire) or locally intensive processes (lake changes and RTS) based on their spatial characteristics and potential impact on permafrost. Although lakes and fire affect a similar fraction of the landscape with 5.04 and 4.75%, respectively, their presence is bound to specific, but mostly opposing, environmental conditions. While lakes dominate in coastal lowlands, wildfires do occur largely in continental inland regions. In contrast, RTS are limited to formerly glaciated surfaces with buried glacial ice or deposits associated with Pleistocene-age syngenetic ice-rich permafrost.

Lakes (118,182 km²) and their changes ($-1737$ km², $+3030$ km², $-4767$ km²) are the most extensive of the evaluated PRDs. They have a potentially severe impact on the ground thermal regime and thermokarst, with lakes raising mean annual ground temperatures above freezing, about 10–15 °C higher compared to tundra in northern Alaska[40]. Lake dynamics are directly linked to the presence of lakes, which cover a larger area than wildfire burn scars. Thermokarst and lake expansion lead to the release of the greenhouse gases (GHG) carbon dioxide or methane[8] due to rapid lake shore erosion[41] or thaw bulb (talik) development underneath lakes[40,42]. As lake growth is a more gradual process on decadal time scales with less variance compared to highly dynamic and variable lake loss that often is characterized by catastrophic events, GHG emissions resulting from lake expansion are potentially more predictable. However, strong precipitation events in 2006 and 2007 in Central Yakutia[16,39] lead to rapid lake expansion in existing thermokarst basins and underline the potential for regionally rapid and dramatic lake area changes in both directions given the right landscape, hydrological, and climatic settings.

In contrast, the net carbon budget of shrinking lakes is highly dependent on the balance between carbon sequestration due to post-drainage permafrost aggradation[43,44] and peat accumulation[45,46] versus formation of methane-emitting wetlands[47]. Our analysis of lake regions spread out over broad environmental gradients in the northern permafrost region revealed that the key drivers of lake change are highly diverse and could not be linked to single mechanisms alone such as permafrost extent, ground-ice or climate. This supports the hypothesis that lake thermokarst is a self-reinforcing process where lakes, once initiated and reaching depths below the maximum lake ice thickness threshold, may further evolve independent of current climatic conditions, making lake area dynamics a difficult-to-interpret indicator of climate change impacts on permafrost regions. Regions with a high abundance of lakes, but with little lake change, may indicate a legacy of past environmental and landscape conditions under which lakes could form and grow rather than present-day conditions.

However, on a regional scale, drivers of lake change can be determined for specific environmental conditions. For example, our observations of increased lake drainage in discontinuous to continuous PF zones in Western Siberia and Alaska may indicate that thermokarst lake changes are related to the transition between different hydrological regimes when permafrost becomes more and more discontinuous. While ground-ice content and geomorphological differences have a strong influence on local and regional-scale lake dynamics[23,48], the detail and quality of globally available datasets of permafrost properties are low. Maps are only available at coarse spatial resolutions that do not resolve local-scale permafrost and ground-ice variation present in many locations such as northern Alaska. This highlights the necessity for improved dataset on permafrost distribution and especially on ground-ice content.

After lakes, wildfires are the PRD with the largest spatial extent of 110,960 km$^2$ in the studied transects of $2.3 \times 10^6$ km$^2$ during the observation period. Generally, fire frequency and extent decreased sharply towards the taiga tundra ecotone, where the density of available fuel decreases and less favorable climatic conditions prevail. Wildfires in the permafrost region have a wide range of impacts on the ground thermal regime, depending on the burn severity and soil moisture conditions[36,49], and have the potential to rapidly cause permafrost degradation[35,50]. Consequently, wildfires may cause other disturbances, including the triggering of thermokarst lake formation[51] or drainage[35] as well as RTS development[35,52,53]. In this study, we did not analyze burn severity, but remote sensing is a viable method to specifically analyze this type of information. Follow-up studies could investigate the specific impact of detected fire events on permafrost stability.

The relatively infrequent occurrence and sparse distribution of arctic tundra fire scars show their currently limited contribution to permafrost degradation. However, tundra fires can lead to local widespread permafrost degradation several years post-fire[35] and with continued warming and shrubification, tundra fires are predicted to increase in frequency[29] and may spread to tundra regions where fires are currently exceptional.

Active RTS are the most localized and smallest PRD as they affected only 5.29 km$^2$ ($n = 408$) of the total study area, several orders of magnitude smaller than the footprint of lakes (118,182 km$^2$), lake changes ($-1737$ km$^2$, $+3030$ km$^2$, $-4767$ km$^2$), or fire scars (110,960 km$^2$). However, they have the most severe and immediate impact on permafrost through their ability to thaw and remove large quantities of ground material within a short time period. RTS impact the ground thermal regime, downstream biogeochemical dynamics, and may decrease terrain stability triggering further mass wasting activity[15]. However, due to the

small footprint of RTS following an exponential distribution with features from few m$^2$ to a maximum of approximately 70 ha for the currently largest known RTS (Batagaika slump)[54], large numbers of particularly smaller features and related processes, such as active-layer detachment slides, may very likely not be sufficiently detectable with 30-m resolution data. Their clustered occurrence within a narrow range of climatological, geo-cryological and topographic conditions, e.g. the abundance of excess ground-ice, the location of former glaciation boundaries, where buried glacial ice may be preserved in sufficiently sloped terrain, can help to make efficient use of very high-resolution imagery to find much smaller RTS and related landscape disturbances.

**Implications for future permafrost.** Once permafrost begins to degrade following a PRD, a range of feedback mechanisms may lead, for example, to changes in the hydrologic and biogeochemical dynamics in the affected area. As the initiation and magnitude of these changes is dependent on surface geology and permafrost characteristics, potential consequences are spatially diverse. While thaw of permafrost in some regions may result in wetting, ponding, and lake formation[39,48], in other regions the opposite may happen with drying of landscapes and drainage of lakes[22,48]. While permafrost thaw itself will generally result in activation of permafrost carbon and its availability for microbial decomposition, the subsequent hydrological conditions have a critical role for the biogeochemical trajectory, in particular an aerobic versus anaerobic decomposition pathway. Depending on these pathways, either carbon dioxide or methane emissions from thawed permafrost following PRDs will dominate the greenhouse gas production in permafrost thaw landforms[8,9]. Similarly, the mobilization and lateral export of permafrost carbon in the form of particulate or dissolved organic carbon will follow different trajectories for these PRDs. As lakes, lake changes, and RTS are inherently linked to permafrost thaw with rising ground temperatures, we generally expect an increase in the areas affected by PRDs in a rapidly warming Arctic and Boreal region, in particular under the business as usual IPCC scenario RCP8.5. The projected increase in wildfire abundance, severity, and return intervals in the warming northern high latitudes[28] likely also favors the increased triggering of permafrost thaw, in particular in regions where permafrost is currently protected by ecosystem characteristics such as thick soil organic layers and vegetation. As a consequence, we expect permafrost to more rapidly degrade across large areas than currently projected by models since these only take into account gradual top-down-thaw without considering PRD-driven thaw.

Our comprehensive overview of the spatial extent and distribution of three key PRDs over about 10% of the northern permafrost region ($2.3 \times 10^6$ km$^2$) provides an unprecedented dataset for a range of potential applications and studies. It allows useful insight into important drivers and their complexity for PRD development when considering very large regions. The massive spatial scale and abundance of some of these processes over a short 16-year observation period, affecting nearly 10% of the studied region, demonstrate the critical need to consider pulse disturbance processes when projecting permafrost dynamics and the future of its carbon pool. So far, land surface schemes in various Earth System models do not include these PRD dynamics largely due to challenges with their implementation on a climate model grid scale. While lake presence, lake changes, wildfire scars, and RTS are among the most important PRDs, other processes such as permafrost peat plateau collapse, ice-wedge degradation, thermo-erosional gully formation, and coastal erosion are important additional disturbances that allow rapid thaw of

permafrost and thus mobilization of permafrost soil carbon. New sensors with a higher spatial-resolution and very short repetition cycles (e.g. Sentinel-2: 10 m, Planet: 3 m) have the potential to overcome the limitations of Landsat and to further enhance the detection limit for smaller features and allow monitoring of a broader variety of disturbances.

The quantitative information from this study on major PRD types may serve as valuable guidance into how these disturbances need to be parameterized in global and regional-scale models to better understand and predict the complexity of landscape processes and biogeochemical cycles in permafrost environments in the present, the past, and the future.

## Methods

**Trend calculation**. We applied trend analyses on all available Landsat (TM, ETM+ and OLI) surface reflectance data of the study regions in a defined range of parameters. Data were preprocessed to surface reflectance and provided by the ESPA processing interface of the United States Geological Survey (USGS) (https://espa.cr.usgs.gov). In order to only capture peak-summer season information and to ensure an acceptable data quality, we used data with land cloud cover of less than 70% and imagery from July and August. We refined the observation period to the years 1999 through 2014, to keep the data amount and quality as consistent as possible, since large parts of Siberia and some coastal regions of Alaska have large gaps in the Landsat archive before 1999 [55]. Overall, we used 14,173 Landsat (T1: $n = 2714$; T2: $n = 3647$; T3: $n = 4947$; T4: $n = 2865$) scenes for the entire analysis with the majority ($n = 12,217$) coming from Landsat Pre-Collection processing. We masked all low-quality pixels, including clouds, cloud shadow and snow, with the FMask layer [56], which is distributed with the data products. Between 5 and 179 valid observations with a mean of 59.3 and standard deviation of 22.0 (T1: $\mu = 49.1$, $\sigma = 16.5$; T2: $\mu = 71.4$, $\sigma = 21.3$; T3: $\mu = 57.9$, $\sigma = 24.0$; T4: $\mu = 58.4$, $\sigma = 15.5$) were recorded for each $30 \times 30$ m each pixel. Locations with less than five observations (e.g. glaciers, aufeis, shadows) were left out from the analysis. We did not apply cross-calibration between different Landsat sensors. Spectral changes of observed PRD strongly exceeded potential minor sensor-calibration inconsistencies. Furthermore, due to the use of the same data source for the analysis and relative comparison against each other, the effect of cross-calibration errors was further minimized.

We calculated six widely used multi-spectral-indices (MSI), Normalized Difference Vegetation Index (NDVI), Normalized Difference Moisture Index (NDMI), Normalized Difference Water Index (NDWI) as well as Tasseled Cap Brightness (TCB), -Greenness (TCG) and -Wetness (TCW), which were chosen to represent a range of different physical surface properties, such as moisture, albedo or vegetation status. For each pixel and MSI, we calculated robust trends based on the Theil−Sen algorithm [57,58], which is more robust against outliers than traditional least-squares regression [59] and has been applied in several remote sensing studies [48,60,61]. The trend analysis returned the slope and intercept parameters, as well as the confidence intervals ($\alpha = 95\%$) of the trend slopes. The scanline corrector (SLC) issue on ETM+ on post-2003 data slightly impacted the trend slope results with a weak striping pattern. Landsat trend data of the four transects are publically available on the PANGAEA data repository at https://doi.org/10.1594/PANGAEA.884137.

**Landscape process classification**. For the detection and delineation of permafrost-related disturbances, we translated the spectral trend information to semantic classes of land and change processes using supervised machine-learning classification (see Supplementary Fig. 3). We applied machine-learning models based on the Random Forest method [62] (RF) with two different configurations, depending on the classification targets; lakes and lake changes (configuration C1) or RTS and fire (configuration C2). Configuration C1 contains four classes, focusing on land−water interactions, with stable classes "land" and "water" as well as dynamic change classes "land to water" and "water to land". C2 contained two additional classes "fire" and "retrogressive thaw slumps" to capture the wider range of land change processes. For the lake change analysis we chose C1 over C2 due to its better accuracy for the detection of land−water change-related processes (see below).

For the training process, we selected 973 point locations of known land cover and land cover change for C1 and 1254 for C2, which are distributed over several areas in the permafrost region (Supplementary Fig. 4, Supplementary Table 1). Due to the spatial heterogeneity and spatial distribution occurrence frequency, a randomized or gridded location selection was not feasible. Therefore, we applied a mixture of random and manual selection of locations of known land cover or changes, based on high-to-moderate resolution imagery (≤30 m, e.g. Worldview, Ikonos, Rapideye, Landsat), in situ observations, aerial survey flights, auxiliary data and Landsat trend data. We calculated the probability values for each of the defined landscape change/no-change classes. The classification model was trained with four calculated trend parameters of the six MSI (see above), the confidence interval range of the trend slopes, terrain elevation and slope as well as latitude and

longitude, in total 34 different attributes, which were calculated for each $30 \times 30$ m pixel. Both classification configurations were fivefold cross-validated. Classification results and single class probabilities were used for later object-based analysis of lakes, wildfires and RTS.

Configuration C1 discriminated the defined classes with very high accuracy (OA: 99.9%, Cohen's kappa: 0.99). Configuration C2 discriminated the defined classes well (OA: 89.0%, Cohen's kappa: 0.85). Classes "land", "land to water" and "water to land" were classified with high accuracies (F-Score > 0.9) with minor class imbalances for water-to-land. The Class RTS was accurately classified (F-Score: 0.86) with few misclassifications to the other dynamic classes. Fire was classified with an F-Score of 0.67 (0.72 recall, 0.64 precision). The majority of misclassifications occurred between fire and the stable land class due to the fuzzy transition between these classes, caused by the presence of old fire scars and a strong variety of fire intensity, vegetation types, and spectral ground signal.

**Extraction of lakes and lake changes**. Lake locations and lake change information were extracted using object-based image analysis and subpixel analysis of machine-learning classified land cover and land cover change probabilities. We used the classification output (class assignment and class probabilities) of C1 to detect and delineate individual lakes.

We extracted individual lake objects as connected pixels of stable water and both change classes (land to water, water to land). The resulting individual lake objects were split into stable and dynamic zones. In the dynamic zone subpixel probabilities of the machine-learning classification were applied to quantify lake changes, whereas the stable water zone was defined as a static water surface.

Lakes smaller than 1 ha were automatically removed from the analysis. Furthermore, we filtered rivers and classification errors from the analysis, using a two-class RF machine-learning classification model using object statistics (mean, standard deviation) of shape- (area, perimeter, orientation, eccentricity, solidity), change (area changes), and auxiliary data (ESA DUE DEM, Global Forest Change [30]). The model was trained by a manually classified dataset of 12,039 potential lake objects, which are distributed over several regions with a binary distinction of "lake" or "no lake". A more detailed description of the lake change analysis is provided in Nitze et al. [48].

**Extraction of wildfire burn scars**. We used the publicly available Global Forest Change (GFC) data [30] in version 1.3 available at https://earthenginepartners.appspot.com/science-2013-global-forest/download_v1.3.html in our assessment of wildfire burn scars. This dataset covers the period from 2000 until 2015, which is shifted by 1 year, compared to the trend analysis, but also covers a 16-year period.

We used the "forest cover loss" data, as a predictor for fire, because wildfires are the nearly exclusive source for forest loss within the study regions. Furthermore, the fire dataset contained several small speckle objects indicating noise and due to their limited size were discarded as non-wildfire.

As the GFC dataset is only sensitive to densely forested area changes, we used the multi-spectral Landsat trend dataset to delineate fires in tundra and to improve burned area perimeters in sparsely forested regions (Supplementary Fig. 5), called trend-based fire mask (TBFM) hereafter. Both datasets, GFC and TBFM, were used complementary to minimize their thematic and spatial limitations, such as the focus of GFC on forest or misclassification of the TBFM with older fire scars (see Landscape process classification).

For the definition of the TBFM we used the classification output (class assignment and class probabilities) of C2 to delineate fires in tundra and to improve burned area perimeters in sparsely forested regions. Pixels with a fire probability of >50% were added to the TBFM. For removing noise and minimizing misclassifications on the pixel level in the GFC and TBFM, we applied several morphological filters where we removed objects smaller than 64 pixels (px) (20 ha), filled holes smaller than 36px and morphologically opened/closed with a round element with a diameter of 5px (150 m) and again removed objects smaller than 20 ha to remove further noise or nonfire forest loss, such as from infrastructure development. Image cleaning operations were carried out using the scikit-image package for the python programming language.

The fire occurrence dataset was compiled from the preprocessed TBFM and GFC datasets. TBFM objects, which intersect GFC objects as well as TBFM objects in tundra were selected and merged with the full GFC dataset to fill omitted burned areas of the GFC data in tundra, forest tundra regions or sparsely forested older burn scars (see Supplementary Fig. 5).

Finally, nonfire forest loss, such as infrastructure development or wood harvest affected areas within the fire occurrence dataset were manually removed based on their shape and vicinity to infrastructure. Western Siberia was strongly influenced by expansive infrastructure development and forestry activity leading to forest removal, which was not caused by fire (6.6% of the fire occurrence dataset). Forestry and mining in Eastern Canada (0.8%) and infrastructure development (new railway line and roads) in Eastern Siberia (0.04%) required postprocessing, whereas the Alaska dataset did not require further postprocessing.

The Alaska fire perimeter dataset, available at https://afsmaps.blm.gov/imf_firehistory/imf.jsp?site=firehistory, was filtered to the observation period and further used for selecting testing and validating correct fire perimeters in tundra and forest tundra regions in Alaska.

For the distinction of tundra and boreal wildfires we used the circumpolar Arctic vegetation map (CAVM)[19]. Tundra extent was calculated from the intersection of transect land areas and CAVM. Fire perimeters intersecting the CAVM footprint were counted as tundra fire, the remaining fire perimeters were defined boreal fires.

**Extraction of retrogressive thaw slumps**. We extracted individual retrogressive thaw slumps segments (RTS) from the classification dataset (C2), where pixel-based probability values of RTS exceeded 30% and extracted the bounding box (bbox) of these segments. A low threshold of 30% was chosen to account for the small size of RTS, which may decrease classification probabilities due to mixed pixels. Final segment boundaries were defined based on a two-class $k$-means clustering algorithm, locally applied on the bbox of each segment, where the class of higher $p$ values was selected as an RTS candidate.

The initial segmentation includes a high false positive rate, which required data filtering. We calculated statistics of RTS-classification $p$ values, slope and spatial shape attributes. Slope values in angular degrees were calculated with the gdaldem software using the 90 m ESA DUE ARCTIC DEM (data resampled to 30 m to match Landsat resolution). We automatically discarded objects with a mean probability of <40% (C2—class RTS). The final RTS selection was carried out manually on the remaining object candidates with the support of Landsat trend data and high-resolution optical imagery, where applicable.

**Processing and software**. The processing chain was developed in the programming language python and available packages of geospatial processing (gdal, rasterio, Fiona, geopandas), numerical analysis and statistics (numpy, scipy, pandas), parallelization (joblib) and machine-learning (scikit-learn). Additionally, external software (gdal utilities) was integrated into the processing chain. Geometric vector operations and post-processing was carried out in QGIS (v.2.18) and ArcMap (v.10.5; ESRI). The processing was parallelized for most working steps, namely data preprocessing, Landsat trend analysis, machine-learning classification and feature extraction.

**Auxiliary information**. We used the permafrost map[20] of the International Permafrost Association (IPA) for the extraction of ice content and permafrost extent. We intersected the centroids of lake objects with permafrost extent polygons for the extraction of permafrost extent and ground-ice class statistics.

We used ERA-Interim Reanalysis data from 1979 to 2014, which were provided by the European Centre for Medium-Range Forecast (ECMWF). We downloaded monthly means of temperatures (id: 130) and total precipitation (id:228). Midday and midnight temperatures were averaged to receive monthly temperatures. Precipitation values of 12 h-periods were summed to monthly totals. For the comparison of time periods data were split into the period from January 1979 through December 1998 and January 1999 through December 2014.

**Code availability**. Custom program code may be provided by Ingmar Nitze upon request. For contact details, please consult the author's details stated above.

## Data availability
The permafrost region disturbance datasets, including lake change information, fire perimeters and RTS perimeters for each transect, are available on the PANGAEA data repository under https://doi.pangaea.de/10.1594/PANGAEA.894755. Input or intermediate data may be provided by Ingmar Nitze upon request.

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

## Acknowledgements

The authors thank the developers of the free and open software used for the data processing and Sebastian Laboor for assisting with the map and graphical design. I.N. and G.G. were supported by the European Research Council (#ERC338335), the European Space Agency (GlobPermafrost), the Networking Fund of the Helmholtz Association (ERC-0013) and the German Ministry for Research and Education (BMBF KoPf). B.M.J. was supported by the National Science Foundation under grant OPP-1806213 and the Joint Fire Science Program under grant 16-1-01-8. V.E.R. was supported by the State of Alaska, by the Russian Science Foundation (project RNF 16-17-00102) and Minobrnauka of the Russian Federation (grant RFMEFI58718X0048, No. 14.587.21.0048).

## Author contributions

I.N. designed the study, developed the data processing chain, carried out analysis and led the manuscript writing. G.G. co-designed the study and contributed to writing and editing the manuscript. B.M.J. was involved in the study design and contributed to writing and editing of the manuscript. V.E.R and J.B. contributed to writing and editing the manuscript and provided important background knowledge from local to pan-arctic scales.

## Additional information

**Competing interests:** The authors declare no competing interests.

