## [Peer Review File · Nature Communications]

Reviewers' comments:

Reviewer #1 (Remarks to the Author):

Review of paper entitled "Remote sensing identifies widespread abundance of recent permafrost region disturbances across the Arctic and Subarctic" submitted to Nature Communications.

This paper communicates a thorough and comprehensive study using basic remote sensing data (Landsat) combined with other products to target the occurrence of physical disturbances in the permafrost regions of the North. The geographical focus is on four latitudinal transects distributed around the circumpolar North from the boreal to the arctic climatic zones.

The paper represents, to the knowledge of this reviewer, the first remotely sensing based comprehensive analysis of these permafrost region disturbances (PRDs) and their dynamics over more than a decade that can be claimed to represent observational evidence covering large scales representative of the circumpolar North as a whole. It is as such an extremely valuable contribution to the science of understanding physical permafrost dynamics as affected by disturbance in a changing climate. It may also serve as a benchmark for the natural dynamics that these processes see over a decadal+ timescale and is as such highly worth publication to a wide audience such as that of Nature Communications.

The concern this reviewer has is merely of an overarching nature.

While the paper as mentioned nicely documents the natural dynamics of the PRDs with its very apparent widely differing spatial variation, the tone of the paper is somewhat twisted towards wording that could suggest that more uniform large scale temporal trends, in the frequency and scale of impact these events have, are shown. This starts in the title which says "...identifies widespread abundance of recent...". This could read as the paper is documenting recent increasing trends in the frequency and scale of PRDs but the documentation only supports the natural spatial variability over the 16 year time slot but not any evidence of changing temporal trends.

Also the "vulnerability of permafrost terrain and carbon pools" (from the abstract) may well be present but this is not really what is documented in this paper. Rather what is shown here is the natural dynamics of PRDs and their spatial distribution against which any future change may be speculated on but again the latter is beyond what the current paper supports. The authors, however, do point at the need for the processes behind the individual PRDs be parameterized for implementation in land surface models to be used in a predictive way. This is entirely a legitimate claim, which the documented paper can form a solid background for, but isn't doing itself. In general this overarching comment relates to an element of the paper risking the criticism to be seeing what it wants to see rather than what is in fact there. The latter is perfectly enough to warrant publication: a clear documentation of natural dynamics in the circumpolar north in relation to PRDs in all their complexity for example causing wetting in some areas and drying in others in response to the same physical forcing. But the paper need to be clear about that there is no documentation of any temporal trends in the frequency of these PRDs and neither any reliable direct forward looking modeling of such.

According to this reviewer, the paper may be published as it stands with a slightly more precise and cautious wording implemented relating to the above mentioned.

Reviewer #2 (Remarks to the Author):

Review, 25 April 2018

Review of the manuscript "Remote sensing identifies widespread abundance of recent permafrost region disturbances across the Arctic and Subarctic" submitted to Nature Communications.

The manuscript identifies the current lack of “spatially and temporally consistent inventories of permafrost affecting local landscape changes across very large regions in sufficiently high spatial resolution” as an important gap in research to be addressed. Corresponding data, with a resolution of 30m, from 1999 to 2015 is presented for four latitudinal transects covering 2.3 Million square kilometers in total. It is derived from time series of remotely sensed data and machine learning. Data are represented in (very beautiful) maps and graphs, overview tables, and a result section that presents major patterns, separated by the type of ‘permafrost region disturbance’ or PRD in the authors’ formulation. This is followed by an interpretation and discussion in the sections named “Implications for Arctic and Boreal Regions” and “Implications for Future Permafrost”.

I am in agreement with the authors that the gap they address is of high importance and I find the information generated will be very valuable. At the same time, I do not find Nature Communications an appropriate outlet for the manuscript presented and recommend rejecting it. In my assessment, the manuscript is not “high-quality research representing an important advance of significance to specialists” as stated on the journal home page. It would be much better suited for a more technical journal with a narrower scope. An important advance requires a more focused approach and methodology for learning.

The current manuscript is motivated mostly by missing data, rather than by a research question that is distilled well based on the state of the art and then addressed with an interpretation of the data presented. Lines 83–212 show results, which are largely a summary of the data, and the subsequent sections on “Implications for Arctic and Boreal Regions” and “Implications for Future Permafrost” are interesting but not suitable to advance our understanding of these systems significantly.

In my mind, the high value of this study would be better materialized by publishing the data, its derivation and testing in more detail and THEN make specific and developed analyses of permafrost systems with it.

Reviewer #3 (Remarks to the Author):

This study used the Landsat satellite image record since 1999 to map the distribution of four types of permafrost-related disturbances (lakes, lake changes, wildfires, and thaw slumps). It also used ancillary spatial data on permafrost type, ground ice, vegetation type (tundra vs forested), and temperature to investigate associations between permafrost condition and disturbance frequency. The paper is well written, the discussion is clearly presented, and supporting figures are excellent.

The major advance and novelty of this work relates to the large spatial extents that were analyzed over four globally distributed transects, which cover 10% of the Northern Hemisphere permafrost region. As emphasized in the manuscript, large-area inventories of permafrost landscape disturbances are not currently available, and this presents a critical limitation for land surface models that aim to represent the permafrost carbon feedback. This research demonstrates that such inventories are now feasible using the freely available 30+ year Landsat archive, combined with computing advances for big data analysis. While the paper demonstrates that thermokarst lake changes and wildfires are widespread across the permafrost zone, this is not a new finding but instead reinforces and scales-up previous studies that were conducted over smaller regions. The analysis of lake changes (growth and shrinkage) across the circumpolar transects with diverse permafrost distribution, ground ice, and geological history also expands on previous regional studies and shows that the direction and intensity of lake area changes is complex and does not follow any simple, consistent pattern.

One aspect of the manuscript that needs improvement is the description of methods, especially the

section dealing with the extraction of wildfire burn scars (lines 367-398), which I found to be confusing in several respects. The addition of one or two figures may help to better convey the workflow used. One limitation of the analysis is that no assessment of change classification accuracy is provided from the random forest classification cross validation (or for the final change maps derived after performing post-processing steps after the random forest classification, which would require holding out some of the reference samples).

Comments on methods:

1. Line 372 - A global forest cover loss product created using Landsat (Global Forest Change or GFC Product) was used as a "predictor" for fire. From the description provided, it is not clear exactly how this GFC product was combined with the random forest classification probabilities that were generated (again, a figure could help here). Was it used to derive training data, and is it suitable to use a generic, Landsat-based global forest change product as the basis for a more refined Landsat product that aims to map fire disturbance in permafrost regions? For example, the GFC may not be optimized to achieve high accuracy for detecting forest loss in sparsely treed boreal forest within the tundra transition zone. Also, why would the GFC product provide a better delineation of forest loss from fire than the method implemented in this study when both use Landsat change metrics and machine learning?
2. Line 389 – Not clear what is meant by the trend-based fire mask was filtered to fire perimeters from the Alaska fire perimeter dataset. Also, does this mean that only the Alaska portion of this one transect was processed in this manner?
3. Line 363 – Would be helpful to provide a sentence or two describing the lake mapping methods implemented in Nitze et al. (2017). How can Landsat trend coefficients be used to effectively map static features like lakes, since most other stable areas would also have no significant trends?
4. It would be useful to know how many Landsat scenes were processed and what computing approach was used (e.g., parallel computing?). Did you use Landsat Collection-1 data?
5. Line 326 – Landsat 8 uses different spectral channels compared to Landsat 5/7. If images from the different sensors are combined, you should explain how this would have a minimal effect on trend results (because NIR reflectance of vegetation is typically higher in L8, this would likely cause L8 TCG values to be systematically higher). The inclusion of L8 imagery should at least be partially mitigated by that fact that relatively large changes (i.e. LC conversions) are being mapped and that surface reflectance data were used, which eliminates the varying effect of atmosphere on the slightly different spectral bands.
6. Line 335 – The mean and SD may be more informative here.
7. Line 348 – Not clear why you used two different random forest training configurations.
8. Line 359 – This appears to sum to 26 features or attributes instead of 30.
9. Line 410 – Unclear what "spectral probability of <0.4%" refers to.

Other comments unrelated to methods

1. Line 137 and 142 – The three numbers cited in parentheses may be confusing because previously it is only the gross increase and decrease presented in parentheses. State that the first number here represents net change.
2. Referring to all lakes as "permafrost disturbances" could be misleading to readers because most lakes would have likely formed 5,000-10,000 years ago during the Holocene climatic optimum (or earlier) and this study uses Landsat trends to map disturbances over a very recent timeframe. Perhaps the term "thermokarst landform" would be preferable for lakes that existed at the beginning of the Landsat study period. Also, as mentioned in the MS, few if any of lakes in transect 4 would be of thermokarst origin so why call them disturbances? (I guess one could argue that they are glacier disturbances).
3. This study addressed some of the major permafrost disturbances that are amenable to mapping using Landsat. However, the discussion cites many other types of disturbance that weren't considered but could be widespread (ice-wedge thermokarst, coastal erosion, collapse scar bogs, and active layer detachment slides). It would be worthwhile briefly discussing the prospects and

requirements to include these features in future satellite mapping studies. For example, could 10 m data like Sentinel-2 overcome the resolution limitation?

4. Did you detect any of the recently documented, new crater lakes on the Yamal Peninsula that are thought be caused by methane explosions?

5. Fig 3. – perhaps the locations of slumps would be more visible if a different colour like blue were used instead of green (especially for T3).

We carefully addressed the reviewer's general and specific comments and concerns and adapted our manuscript accordingly. We slightly changed the title and applied minor changes to the main body of the manuscript. The methods section was expanded significantly upon recommendation of reviewer #3. Based on this reviewer's comments we slightly updated our fire detection workflow, which resulted in minor changes of wildfire quantifications. Furthermore, we added two supplementary figures and more detailed descriptions of specific methods. Please find our detailed responses to the reviewer's comments and remarks below in red color.

Furthermore, our datasets are now available for the reviewers under following links:

Lakes: <https://1drv.ms/u/s!AobXXrP933xWpBhujXf0ljKq1oaC>

Fire: <https://1drv.ms/u/s!AobXXrP933xWpBmxKu-GrhsQ5baB>

Retrogressive Thaw Slumps: https://1drv.ms/u/s!AobXXrP933xWpBokpx7r_gH_52Z

We thank the reviewers for carefully assessing our manuscript.

Sincerely,

The author team.

Reviewer #1:

Review of paper entitled "Remote sensing identifies widespread abundance of recent permafrost region disturbances across the Arctic and Subarctic" submitted to Nature Communications.

This paper communicates a thorough and comprehensive study using basic remote sensing data (Landsat) combined with other products to target the occurrence of physical disturbances in the permafrost regions of the North. The geographical focus is on four latitudinal transects distributed around the circumpolar North from the boreal to the arctic climatic zones.

The paper represents, to the knowledge of this reviewer, the first remotely sensing based comprehensive analysis of these permafrost region disturbances (PRDs) and their dynamics over more than a decade that can be claimed to represent observational evidence covering large scales representative of the circumpolar North as a whole. It is as such an extremely valuable contribution to the science of understanding physical permafrost dynamics as affected by disturbance in a changing climate. It may also serve as a benchmark for the natural dynamics that these processes see over a decadal+ timescale and is as such highly worth publication to a wide audience such as that of Nature Communications.

The concern this reviewer has is merely of an overarching nature.

While the paper as mentioned nicely documents the natural dynamics of the PRDs with its very apparent widely differing spatial variation, the tone of the paper is somewhat twisted towards wording that could suggest that more uniform large scale temporal trends, in the frequency and scale of impact these events have, are shown. This starts in the title which says "...identifies widespread abundance of

recent...”. This could read as the paper is documenting recent increasing trends in the frequency and scale of PRDs but the documentation only supports the natural spatial variability over the 16 year time slot but not any evidence of changing temporal trends.

We agree with the reviewer that the wording may have unintentionally suggested changes in the dynamics of PRD in different cases. Where appropriate (line 61 + Title) we slightly changed the wording and adapted the title to “Remote sensing quantifies widespread abundance of permafrost region disturbances across the Arctic and Subarctic”.

Also the “vulnerability of permafrost terrain and carbon pools” (from the abstract) may well be present but this is not really what is documented in this paper. Rather what is shown here is the natural dynamics of PRDs and their spatial distribution against which any future change may be speculated on but again the latter is beyond what the current paper supports. The authors, however, do point at the need for the processes behind the individual PRDs be parameterized for implementation in land surface models to be used in a predictive way. This is entirely a legitimate claim, which the documented paper can form a solid background for, but isn’t doing itself.

The reviewer raised a good point here. As we do not focus on carbon vulnerability per se, but rather the processes that potentially affect carbon dynamics, we adapted the wording slightly (abstract-lines 13 and 14). However, we kept the relation to carbon dynamics in the discussion (**Implications for Arctic and Boreal Regions; Implications for Future Permafrost**), as the carbon dynamics play an important role for the motivation of monitoring PRD and are a direct consequence of these disturbances.

In general this overarching comment relates to an element of the paper risking the criticism to be seeing what it wants to see rather than what is in fact there. The latter is perfectly enough to warrant publication: a clear documentation of natural dynamics in the circumpolar north in relation to PRDs in all their complexity for example causing wetting in some areas and drying in others in response to the same physical forcing. But the paper need to be clear about that there is no documentation of any temporal trends in the frequency of these PRDs and neither any reliable direct forward looking modeling of such.

We thank reviewer #1 for these valuable comments and agree that there is no trend towards more PRDs observable for such a short time period. The study emphasizes the general widespread distribution and highly dynamics nature of the observed PRDs, rather than actual trends (change in abundance or intensity), which would require longer-term observation or modeling schemes. The confusion here may origin from the frequent use of the term “trend” the context of describing the MSI trajectories (e.g., an observed increase in MSI constitutes a spectral trend used to detect and characterize a disturbance, but it does not imply a trend of increasing abundance or intensity of that disturbance). We, emphasized the wording towards dynamics of changes. We slightly adapted the title for more clarity and changed other few parts for better clarification (e.g. line 61).

Reviewer #2 (Remarks to the Author):

Review of the manuscript “Remote sensing identifies widespread abundance of recent permafrost

region disturbances across the Arctic and Subarctic” submitted to Nature Communications.

The manuscript identifies the current lack of “spatially and temporally consistent inventories of permafrost affecting local landscape changes across very large regions in sufficiently high spatial resolution” as an important gap in research to be addressed. Corresponding data, with a resolution of 30m, from 1999 to 2015 is presented for four latitudinal transects covering 2.3 Million square kilometers in total. It is derived from time series of remotely sensed data and machine learning. Data are represented in (very beautiful) maps and graphs, overview tables, and a result section that presents major patterns, separated by the type of ‘permafrost region disturbance’ or PRD in the authors’ formulation. This is followed by an interpretation and discussion in the sections named “Implications for Arctic and Boreal Regions” and “Implications for Future Permafrost”.

Point 1 - I am in agreement with the authors that the gap they address is of high importance and I find the information generated will be very valuable. At the same time, I do not find Nature Communications an appropriate outlet for the manuscript presented and recommend rejecting it. In my assessment, the manuscript is not “high-quality research representing an important advance of significance to specialists” as stated on the journal home page. It would be much better suited for a more technical journal with a narrower scope. An important advance requires a more focused approach and methodology for learning.

In our view, the manuscript presents an unprecedented dataset on the abundance and dynamics of PRDs on continental scales and thus highly relevant results, which are of importance for scientists with a focus on permafrost but also carbon cycling. So far, studies quantifying permafrost region disturbances were either locally focused, but hardly comparable across regions due to usage of different data and/or methodologies, or regional-to-pan-arctic focused, but with insufficient spatial resolution to capture local-scale changes.

With this study we aimed to resolve this scaling issue and past inconsistencies, producing new datasets that can be easily assessed by users, and discussed the implications for various study fields, most notably permafrost and future carbon cycling. We argue that this study, which goes beyond previous studies in spatial detail, site extent and particularly data consistency, provides new insights and consistent quantification of PRD, which is of high relevance for research in the northern permafrost region and all its related scientific fields. We therefore see Nature Communications as an appropriate journal for our manuscript.

Point 2 - The current manuscript is motivated mostly by missing data, rather than by a research question that is distilled well based on the state of the art and then addressed with an interpretation of the data presented. Lines 83–212 show results, which are largely a summary of the data, and the subsequent sections on “Implications for Arctic and Boreal Regions” and “Implications for Future Permafrost” are interesting but not suitable to advance our understanding of these systems significantly.

We are aware that the manuscript itself built upon the significant lack of quantifiable data about PRD, which were mostly limited by the availability of data and/or processing techniques. We argue that the overarching research question “How extensive are recent PRDs and do they vary by permafrost extent

and characteristics across the Arctic and Subarctic?” has always persisted, but never been successfully tackled before, due to limitations in data availability and technical processing possibilities. Building upon new technical capabilities and methods this manuscript has a strong technical focus. However, we present the results and discussion in a way to cater for a wide range of potential users and scientific fields.

We added the scientific question to our manuscript “How extensive are recent PRDs and do they vary by permafrost extent and characteristics across the Arctic and Subarctic?” (lines 68/69)

Point 3 - In my mind, the high value of this study would be better materialized by publishing the data, its derivation and testing in more detail and THEN make specific and developed analyses of permafrost systems with it.

We see this comment as a valid point. However, we will publish our datasets along with the manuscript on a public open access data platform (e.g. PANGAEA). In our manuscript we presented the results of these data and their implications. We foresee a further use of these datasets by other users for a variety of applications, such carbon flux calculations, ground-ice prediction, and permafrost modeling from local to global scales among many other possible applications.

Reviewer #3:

1.Line 372 - A global forest cover loss product created using Landsat (Global Forest Change or GFC Product) was used as a “predictor” for fire. From the description provided, it is not clear exactly how this GFC product was combined with the random forest classification probabilities that were generated (again, a figure could help here). Was it used to derive training data, and is it suitable to use a generic, Landsat-based global forest change product as the basis for a more refined Landsat product that aims to map fire disturbance in permafrost regions? For example, the GFC may not be optimized to achieve high accuracy for detecting forest loss in sparsely treed boreal forest within the tundra transition zone. Also, why would the GFC product provide a better delineation of forest loss from fire than the method implemented in this study when both use Landsat change metrics and machine learning?

The limitations of the GFC product occurs in sparsely or non-forested (i.e. taiga to tundra) regions, but performs well in densely forested areas (e.g. boreal forest). This facilitated the need for a fire dataset based on Landsat trends and machine-learning (Trend Based Fire Map: TBFM). The machine-learning classification process (Configuration C2) included fire and was optimized/trained for the entire ecological gradient (Tundra to Boreal forest) and to cover different geological conditions (which impact the spectral signal), but also broaden the semantic detail (e.g. water, erosion/RTS).

The distinction of “stable land” to “fire” proved to be quite fuzzy, as most boreal forest sites are subject to some level of change due to past fire activity, particularly in areas with very strong fire activity. Therefore, the GFC is likely the better dataset in the boreal zone observing fires directly, but fades quickly with decreasing forest densities, such as forest tundra or young forests.

We added two supplementary figures to better demonstrate the workflow (Suppl. Fig.3) and to better visualize the different fire masks and their advantages and disadvantages (Suppl. Fig.5)

2. Line 389 – Not clear what is meant by the trend-based fire mask was filtered to fire perimeters from the Alaska fire perimeter dataset. Also, does this mean that only the Alaska portion of this one transect was processed in this manner?

We used the Alaska fire perimeter dataset for testing and validating the fire mapping in Alaska and transferred the methodology to the other transects. We provide a more detailed and especially clarified description in the manuscript (lines 434 ff.).

3. Line 363 – Would be helpful to provide a sentence or two describing the lake mapping methods implemented in Nitze et al. (2017). How can Landsat trend coefficients be used to effectively map static features like lakes, since most other stable areas would also have no significant trends?

With the inclusion of the *intercept* values of each index, static features (stable land and water) are easily distinguished by these values, as they vary depending on the land surface and basically work like a multi-spectral index of a single observation. In general, the index slopes predominantly determine the change versus no change and type of change distinction, whereas the intercept predominantly determines the type of static surface features.

We added a more detailed description of the general workflow in the lake change detection section to clarify this part. A visual representation of the workflow (flowchart) was added to the supplementary material (Suppl. Fig. 3)

4. It would be useful to know how many Landsat scenes were processed and what computing approach was used (e.g., parallel computing?). Did you use Landsat Collection-1 data?

Most of the Landsat data are of Pre-Collection origin. The data collection/download overlapped the Pre-Collection to Collection-1 Phase. Overall, we used 14,173 Landsat scenes (T1: 2,714; T2: 3,647; T3: 4,947; T4: 2865) for the entire analysis. The majority of scenes are from Pre-Collection Data (n=12,217). We did not notice specific issues arising from the use of both data pre-processing versions. We adapted our processing chain to support both versions.

We have added more detail to the manuscript (lines 337 ff.).

5. Line 326 – Landsat 8 uses different spectral channels compared to Landsat 5/7. If images from the different sensors are combined, you should explain how this would have a minimal effect on trend results (because NIR reflectance of vegetation is typically higher in L8, this would likely cause L8 TCG values to be systematically higher). The inclusion of L8 imagery should at least be partially mitigated by that fact that relatively large changes (i.e. LC conversions) are being mapped and that surface reflectance data were used, which eliminates the varying effect of atmosphere on the slightly different spectral bands.

We are aware of minor calibration differences of the Landsat sensors. We did not apply a between-sensor calibration. The LC changes associated with the observed PRD are rather strong and outweigh any potential calibration errors/noise between MSI for Landsat 5, 7, and 8. Furthermore, the same type

of input data was used for all sites, and in conjunction with the trend analysis and machine-learning classification, absolute calibration errors are further minimized as the data are compared relatively to one other.

We added detailed information to lines 344 ff. in the manuscript.

6. Line 335 – The mean and SD may be more informative here.

Overall, we have 59.3 mean observations per pixel, with some variation between the individual transects. We added this information for the entire dataset and for each transect to the manuscript (lines 341 ff).

7. Line 348 – Not clear why you used two different random forest training configurations.

We used two configurations which are individually specialized for specific applications. Configuration 1 (C1) only included lake related classes and was optimized for lake changes in order to achieve a better tuning of p-values and more accurate lake change statistics, particularly for the sub-pixel level. C2 was used to capture a broader range of PRD, but lacked the high overall accuracy of C1. However, the tuning of p-values was not as critical as for the lake change detection.

We added a more detailed description of the reasoning for the choice of two different classification settings in the manuscript (lines 360 ff.).

8. Line 359 – This appears to sum to 26 features or attributes instead of 30.

The reviewer is correct and very attentive. After checking the classification model and training data we noted the inclusion of latitude and longitude information in addition to the named attributes, in total 32 attributes. We have updated this information in the manuscript.

9. Line 410 – Unclear what “spectral probability of <0.4%” refers to.

This has been changed to <40%. Here we mixed up scales of 0-1 and 0-100%. We have updated this information in the manuscript.

Other comments unrelated to methods

1. Line 137 and 142 – The three numbers cited in parentheses may be confusing because previously it is only the gross increase and decrease presented in parentheses. State that the first number here represents net change.

The structure of lake area change numbers has been adapted to the structure of the other paragraphs and the reviewer’s suggestion. Thank you for this recommendation.

2. Referring to all lakes as “permafrost disturbances” could be misleading to readers because most lakes

would have likely formed 5,000-10,000 years ago during the Holocene climatic optimum (or earlier) and this study uses Landsat trends to map disturbances over a very recent timeframe. Perhaps the term “thermokarst landform” would be preferable for lakes that existed at the beginning of the Landsat study period. Also, as mentioned in the MS, few if any of lakes in transect 4 would be of thermokarst origin so why call them disturbances? (I guess one could argue that they are glacier disturbances).

It is true that many northern lakes formed prior to the Landsat record. However, their presence alone (not including dynamics), has a strong influence on the ground-thermal regime and can be considered a disturbance with of course different implications depending on the ground conditions. As lake changes (gain and loss) are inseparable from lakes, we chose to include them into the same category.

We discussed the role of lakes (and their changes) in lines 223 ff. and their direct impact on underlying and surrounding permafrost. For lakes located in transect 4 we have noted that these lakes are likely not of thermokarst origin. However, we do still maintain the point of view that lakes occupying antecedent landscape depressions still affect the ground thermal regime of the underlying permafrost, even if that permafrost is thaw stable. Furthermore, the comparison of bedrock and non-bedrock sites provides a good perspective or benchmark on the influence of the underlying substrate and ground-ice content on lake dynamics and therefore all associated implications.

3. This study addressed some of the major permafrost disturbances that are amenable to mapping using Landsat. However, the discussion cites many other types of disturbance that weren't considered but could be widespread (ice-wedge thermokarst, coastal erosion, collapse scar bogs, and active layer detachment slides). It would be worthwhile briefly discussing the prospects and requirements to include these features in future satellite mapping studies. For example, could 10 m data like Sentinel-2 overcome the resolution limitation?

With a 10-m spatial resolution, Sentinel-2 is likely be able to detect some of the spatially less extensive disturbances/changes and is a big step forward from Landsat given the resolution is 9 times finer (10x10 vs 30x30 m). With growing archives (2016 and later), S-2 will become more and more important for monitoring applications.

We added a sentence to discuss the potential use of Sentinel-2 and its advantages (lines 320 ff.).

4. Did you detect any of the recently documented, new crater lakes on the Yamal Peninsula that are thought be caused by methane explosions?

The crater (lakes) are typically very small and below the size of our minimum mapping unit. Just out of curiosity we have looked at the Landsat trend data for these sites and some of these are apparent, but in sticking with the robustness of our methods we have not included them in this manuscript. Higher resolution data (Sentinel-2), Planet, or others are more likely to be a useful source to automatically track these features. These interesting features are definitely worth exploring and will surely be an important

target for further analysis. Future versions of our processing chain (or others), using data with higher spatial resolution, may have a great potential to detect and monitor these features.

We refrained from mentioning the emission craters, as they are very likely a spatially restricted feature and not directly in the scope of this manuscript.

5. Fig 3. – perhaps the locations of slumps would be more visible if a different colour like blue were used instead of green (especially for T3).

We changed RTS signature to blue crosses and changed the LGM extent to grey, which makes the figure (Fig.3) better readable.

We thank all reviewers for their critical dissemination of our manuscript and for providing feedback, which helped to improve his manuscript.

REVIEWERS' COMMENTS:

Reviewer #3 (Remarks to the Author):

I have reviewed the authors' response to my review and the revisions to the paper. I am satisfied how they have addressed my comments.